# Transcriptomic analyses of the termite, *Cryptotermes secundus*, reveal a gene network underlying a long lifespan and high fecundity

Silu Lin [1,2], Jana Werle[1,2] & Judith Korb [1✉]

Organisms are typically characterized by a trade-off between fecundity and longevity. Notable exceptions are social insects. In insect colonies, the reproducing caste (queens) outlive their non-reproducing nestmate workers by orders of magnitude and realize fecundities and lifespans unparalleled among insects. How this is achieved is not understood. Here, we identified a single module of co-expressed genes that characterized queens in the termite species *Cryptotermes secundus*. It encompassed genes from all essential pathways known to be involved in life-history regulation in solitary model organisms. By manipulating its endocrine component, we tested the recent hypothesis that re-wiring along the nutrient-sensing/endocrine/fecundity axis can account for the reversal of the fecundity/longevity trade-off in social insect queens. Our data from termites do not support this hypothesis. However, they revealed striking links to social communication that offer new avenues to understand the re-modelling of the fecundity/longevity trade-off in social insects.

[1] Evolutionary Biology and Ecology, University of Freiburg, Freiburg, Germany. [2] These authors contributed equally: Silu Lin, Jana Werle.
✉email: judith.korb@biologie.uni-freiburg.de

L ive fast, die young. This notion points to the fundamental life-history trade-off between fecundity and longevity that characterise organisms and constrains the lifetime reproductive success of animals[1]. A notable exception is social insects (termites, ants, and some bees and wasps). Their queens often produce hundreds of thousands of offspring and can live for decades, compared to a lifespan of 0.1 ± 0.2 years that is typical for solitary insects[2].

At the mechanistic level, the fecundity/longevity trade-off has been associated with evolutionarily conserved molecular pathways that are broadly shared across animals, including the nematode *Caenorhabditis elegans*, the fruit fly *Drosophila melanogaster*, mice, and humans[3–7]. In insects, these pathways were recently dubbed the IIS-JH-Vg/YP circuit (insulin/insulin-like-growth factor 1 signaling–juvenile hormone–vitellogenin/yolk protein)[8]. In short, the nutrient-sensing IIS and TOR pathways have emerged as central regulators of lifespan and fecundity that interact with neuroendocrine JH signaling and regulate, for instance, fecundity-related vitellogenesis[8,9] (Fig. 1). Vitellogenin is well known as yolk precursor necessary for egg production. Generally, in nonsocial insects like *D. melanogaster*, an upregulation of IIS signaling results in high JH titers, required for activating *Vg* expression (or its fly equivalent *YP* expression) (Fig. 1a). However, JH is highly pleiotropic with many life-shortening consequences[10]. A downregulation of the IIS-JH-Vg/YP circuit fosters maintenance- and survival-related processes at the expense of fecundity[8,11]. Thus, this network seems to directly moderate the fecundity/longevity trade-off.

Social insects represent a major exception to the typical fecundity/longevity trade-off. They live in colonies with reproductive division of labor. A hallmark of the female reproductive individuals in these colonies, the queens, is their high fecundity (some termite queens can lay 20,000 eggs per day) which, unexpectedly, goes along with extraordinary lifespans[2,12,13]. This apparent overcoming of the fecundity/longevity trade-off has made social insect queens some of the most fecund terrestrial animals, which helps to explain the evolutionary and ecological success of these taxa[14].

Social insect queens are characterized by a unique suite of traits that distinguish them both from their nonreproductive nestmates and from solitary reproductive females. These include fundamental life-history traits like fecundity and longevity, as well as other important traits involved in the regulation of the reproductive division of labor. To uphold reproductive monopoly, most queens inhibit the reproduction of female nestmates, which perform nonreproductive tasks such as foraging and brood care and which have shorter lifespans. Inhibition of reproduction in workers is especially important in species with workers that retain the capability to reproduce. They make up most of the social Hymenoptera and all termites, except for one family (the Termitidae)[15]. Reproductive dominance is often achieved via chemical communication through cuticular hydrocarbons (CHCs) (though other compounds can also be involved)[16,17]. Queen CHCs have been proposed to function as fertility signals across social insects to affect the behavior and physiology of colony members[18–23].

We refer to the combination of traits that distinguish queens both from their nonreproductive worker nestmates and from solitary reproductive females as the "hallmarks of queenness." Identifying the molecular architecture that underlies the hallmarks of queenness sheds light on essential features of social insects' success and the apparent overcoming of the fecundity/longevity trade-off. Importantly, within a colony, queens and workers are encoded by a single genome, with different phenotypes arising via differential gene expression. Transcriptome analyses are consequently an ideal tool to decode the molecular underpinnings of what makes a queen and the defying of a fundamental life-history trade-off.

What is known about these molecular mechanisms? Studies on bees, ants, and wasps suggest that the IIS-JH-Vg/YP circuit is important to fecundity and longevity regulation, as in solitary insects (e.g., refs. [9,24–29], for recent reviews see: refs. [30–32]). In ant species, queen development has been associated with increased IIS signaling, which often appears to affect vitellogenesis by promoting downstream production of JH (e.g., refs. [25,27]). However, studies of some social Hymenoptera imply that the

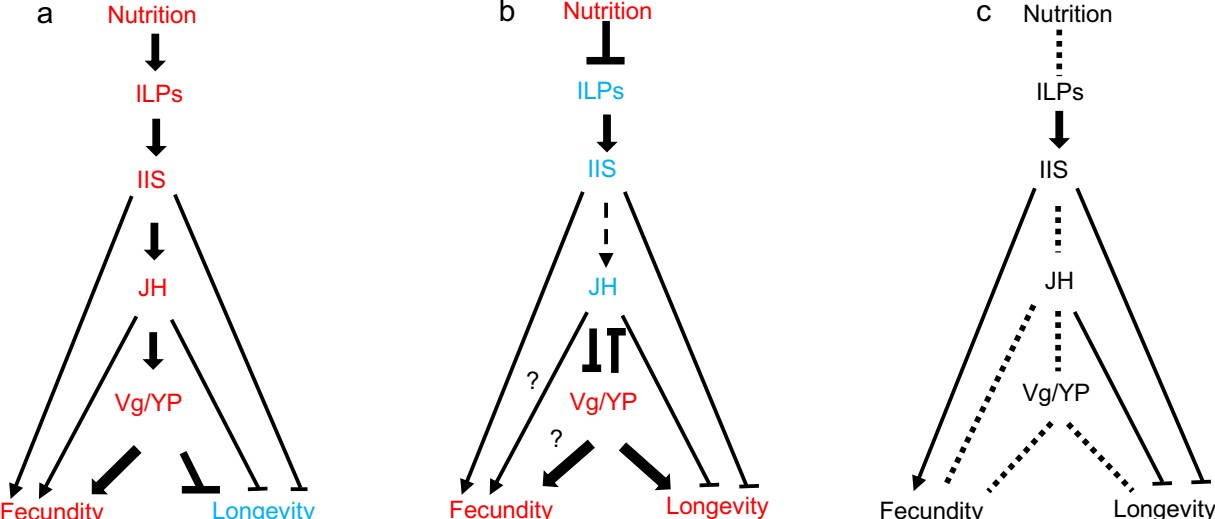

**Fig. 1 Models of IIS-JH-Vg/YP circuit to explain the fecundity–longevity trade-off and its reversal in social insects.** Although the TOR pathway appears to play a role in fecundity and longevity of social insects, it is not addressed in the "rewiring hypothesis" of the IIS-JH-Vg/YP circuit and therefore not included in this figure. **a** The typical circuit as revealed for solitary insects like the fruit fly *Drosophila melanogaster*, **b** a remolded circuit as indicated by results for honeybee queens, *Apis mellifera*, and **c** a rewired circuit as hypothesized for social insects. Adapted from Rodrigues and Flatt[8] and Corona et al.[9]. Solid and dashed arrows represent direct versus indirect activation, respectively, question marks indicate unclear relationships, stop bars indicate repression, dashed lines indicate potential rewiring points, red indicates upregulation, and blue indicates downregulation. For a more detailed explanation, see main text.

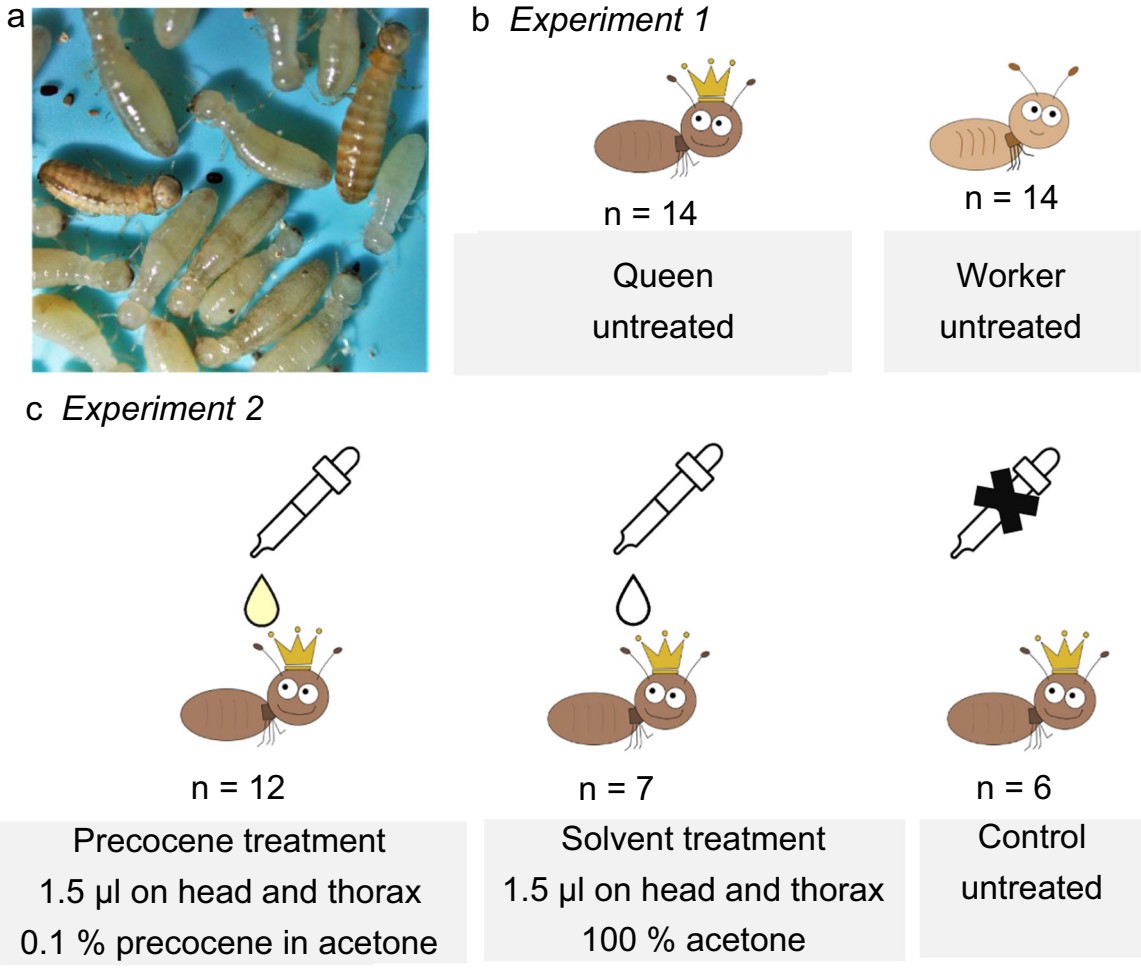

**Fig. 2 Experimental setup. a** Insight into a *Cryptotermes secundus* colony with neotenic queen and king (darker) and workers (paler). **b** Experiment 1: the Queen network experiment compared gene expression between queens versus workers. It included 14 untreated queens and 14 untreated workers. **c** Experiment 2: the JH-manipulation experiment compared gene expression between precocene-treated, solvent-treated, and untreated queens. For the latter, we pharmacologically reduced JH in twelve queens by topical application of 1.5 µl of 0.1% precocene in acetone on head and thorax ("precocene treatment"). To control for solvent-induced effects, we topically applied 1.5 µl of pure acetone to seven queens ("solvent treatment"). As a treatment control we used six completely untreated queens ("control").

fecundity/longevity trade-off can be broken by changes in the regulatory architecture of the IIS-JH-Vg/YP circuit[8,9] (Fig. 1b). First, IIS signaling seems downregulated in old *Apis mellifera* queens compared to old workers[9]. Second, in *A. mellifera*, JH and Vg titers are negatively correlated and constant vitellogenesis does not require concurrently high JH titers[33–36]. Similarly, the reproducing queens of some ants (e.g., *Diacamma*:[37], *Streblognathus peetersi*:[38]) are not characterized by high JH titers. Thus, life-shortening effects of JH are avoided. Last, Vg appears to have multiple roles in the honeybee, including antiaging effects (e.g., refs. [39,40] and references therein). Consequently, it was hypothesized that evolutionary rewiring of the conserved IIS-JH-Vg/YP circuit could explain the co-occurrence of long lifespan and enormous fecundity in social insect queens[8,9] (Fig. 1c).

For termites, much less is known about the molecular underpinnings of queen phenotypes. Termites belong to the cockroach clade (Blattodea) and evolved reproductive division of labor independently from social Hymenoptera[41,42]. Few genetic studies addressed the hallmarks of queenness in termites and such studies have mostly focused on a small set of specific genes and did not perform network analyses[20,43–47]. A comprehensive understanding of the complete IIS-JH-Vg/YP circuit is consequently missing[8,9].

Despite considerable progress in social Hymenoptera[9,24,28,29] no study could identify a comprehensive gene network that characterizes reproducing queens. Such a central gene network, unique to queens, must include mechanisms for prolonged lifespan and high fecundity, as well as such genes underlying reproductive division of labor and chemical inhibition of worker reproduction.

Here, we aimed to identify gene networks characterizing queens and the reversal of the fecundity/longevity trade-off in termites. We analyzed gene expression differences between queens and workers in the wood-dwelling termite *Cryptotermes secundus* (Fig. 2a) using head–prothorax transcriptomes of 28 individuals (queen network experiment) (Fig. 2b; sample details in Supplementary Data 1). Next, we pharmacologically reduced JH with topical application of precocene solutions (including a solvent control) to test how low JH titers affect network connections (JH-manipulation experiment) (Fig. 2c) and especially, whether a rewiring of the IIS-JH-Vg/YP circuit exists. For this, we used a total of 25 head–thorax transcriptomes (sample details: Supplementary Data 1). For both experiments, we first performed differential gene expression analyses. Second, we applied weighted gene co-expression network analyses (WGCNAs) using all genes to identify and characterize modules of co-expressed genes (i)

that were positively associated with queens (queen network experiment) and (ii) that were associated with different pharmacological treatments (JH-manipulation experiment). For the JH-manipulation experiment, we compared (i) precocene treatment versus untreated control to investigate a JH effect that might be confounded by a solvent effect, (ii) solvent control treatment versus untreated control to investigated only the solvent effect, and (iii) precocene treatments versus solvent control treatment to identify the "pure" JH effect. The solvent–control treatment is necessary to disentangle the JH reduction effect from handling artifacts as the solvent alone, acetone, can have effects[48]. For all identified modules, we also did KEGG enrichment analyses. WGCNA identifies modules of co-expressed genes that are associated with a trait (e.g., queen/worker; precocene/control) and determines the significance of the module–trait and gene–trait associations. Head–prothorax transcriptomes appear well suited for our purpose as they likely encompass components from neuroendocrine signaling (e.g., JH biosynthesis by the corpora allata), chemical perception and production (antenna and prothorax, respectively), as well as fat body (prothorax)-related vitellogenesis. We might have missed stronger signals from the abdomen or the ovaries (see also "Results and discussion"). Our results provide fundamental insights into the enigma of defying life-history trade-offs, the basis of social life in insects.

## Results and discussion

**Queen network experiment.** Analyses of 28 transcriptomes (14 queens and 14 workers) revealed 1811 significantly differentially expressed genes (DEGs) between queens and workers. Of these, 736 DEGs were upregulated in queens, while 1075 DEGs were upregulated in workers (Supplementary Data 2A).

Applying WGCNA to all expressed genes (not only DEGs) identified 176 modules of co-expressed genes that we named provisionally after colors assigned by the WGCNA package. Twenty-one modules were positively associated with queens while 22 modules were positively associated with workers (Supplementary Data 3A). To identify modules that characterize queen phenotypes, we manually explored gene modules that were positively associated with queens and that contained *C. secundus* genes annotated as chemical communication genes and TI-J-LiFe (TOR/IIS-JH-Lifespan and Fecundity) genes (see "Methods"; for gene list see Supplementary Data 4). The TI-J-LiFe gene list is a list of genes from the TI-J-LiFe network that combines all major pathways underlying aging and life-history trade-offs, including the IIS-JH-Vg/YP circuit. This list was originally generated for *D. melanogaster*[10,40,49–52]. Our search resulted in seven interesting queen modules (turquoise, brown1, blue2, salmon2, sienna4, coral4, and tan4; hereafter "queen modules") (Supplementary Data 3A). Most striking were the queen modules turquoise, brown1, and blue2. The turquoise module encompassed genes from all essential TI-J-LiFe pathways (except for TOR) and we coined the term "queen central module" (QCM). The queen modules brown1 and blue2 included many genes related to the production and recognition of CHCs, respectively. For results from KEGG enrichment analyses see Supplementary Data 5.

The QCM had a significant positive association with queens ($P < 0.001$), separating *C. secundus* queens from workers. It comprised genes from the IIS-JH-Vg/YP circuit as well as genes associated with carbohydrate metabolism, neuro-hormonal signaling, and CHC biosynthesis (Supplementary Data 3A). The hub gene of the QCM was *Csec-Cytb5-r-a*, which encodes a cytochrome b5-related protein. The following genes were significantly positively associated with queens ($P < 0.05$) or by trend ($P = 0.05$–$0.09$) (Figs. 3 and 4). The pattern of co-expressed genes indicates that we have uncovered a central module linking

all major molecular pathways underlying life-history regulation and the hallmarks of queenness (Fig. 5).

Concerning JH signaling, the QCM contained several JH-related genes (Fig. 3). Their co-expression pattern implies high JH titers. The presence of the female JH epoxidase gene of *C. secundus* (one of two *JH epoxidase* genes occurring in *C. secundus*[53,54]) indicates upregulation of JH biosynthesis, as its enzyme catalyzes the last step of JH production in the corpora allata of insects. This is supported by available results of quantitative real-time PCR (qRT-PCR) experiments, which show an overexpression in queens compared to workers. The notion that the QCM points to high JH titers in queens was further supported by an upregulation of several genes characterized as *takeout*: *takeout-3, takeout-4,* and *takeout-9*. *Takeout* genes encode JH binding proteins (JHBP). JH binds to JHBPs during transport in the hemolymph to target tissues. Finally, a queen-specific upregulation of the JH early-response gene *Kr-h1* (*Kruppel homolog 1*) in *C. secundus, Csec-Kr-h1*, from another queen module (queen module tan4) confirmed high JH activity in *C. secundus* queens. An overexpression of *Csec-Kr-h1* in queens compared to workers is confirmed by qRT-PCR data.

In line with this signature of high JH titers, many fecundity-related genes were co-expressed in the QCM (Fig. 3). They included the *C. secundus* vitellogenin gene *Neofem3*[43] as well as the two other vitellogenin genes (*Csec-Vg-1* and *Csec-Vg-2*) occurring in *C. secundus*[53]. Vitellogenins are storage proteins and yolk-precursors that are produced in the fat body and then carried via the hemolymph to be deposited into the developing egg (e.g., ref.[55]). Vg (in the honeybee and ants) as well as Vg-like proteins (in ants) have also been associated with behavioral maturation and task division among workers (e.g., refs.[56–59]). The three queen-associated vitellogenin genes we found encode conventional Vgs. Conventional Vgs are the only Vgs found in cockroaches, which include termites, with a termite-specific Vg duplication[53]. Although they may have other functions as well (for instance, as antioxidants), at least *Neofem3* seems to be directly related to reproduction in *C. secundus*[43] and *Cryptotermes cynocephalus*[44]. This was confirmed with qRT-PCR results in both studies. Further, in the dampwood termite *Zootermopsis nevadensis* all three Vgs are upregulated in queens (and partly in kings) compared to all other castes, with the strongest upregulation of the *Neofem3* ortholog[60]. Currently, not much evidence exists that supports subfunctionalisation of Vgs in termites, though further functional investigations are required. The co-expression of these vitellogenin genes in the QCM was complemented by another putatively egg-related gene, *Csec_G13850*, which was characterized by a lipovitellin–phosvitin complex.

We also detected a strong carbohydrate-related signature in the QCM (Fig. 4). Ten *trehalose transporter* genes (*Tret*) were co-expressed in the QCM. The disaccharide trehalose is a major sugar in insects[61]. It is produced in the fat body from glycogen and circulating in the hemolymph transported by Trets[62]. Furthermore, we detected genes related to the regulation of metabolism in queens. First, *Csec-Tobi* (target of brain insulin; part of queen module coral4) was positively associated with queens. *Tobi* encodes a glycogen-specific glucosidase that promotes glycogenolysis in insects[63]. Second, two putative AKH/RPCH (adipokinetic hormone/red pigment-concentrating hormone)-related genes, *Csec_G14435* and *Csec_G12895*, were co-expressed in the QCM. The gene *Csec_G14435* is closely related to the hypertrehalosemic hormone (HTH) receptor of the cockroaches *Blattella germanica* and *Blaberus discoidalis*. In *B. discoidalis*, JH increases fat body protein synthesis, modulated and amplified via HTH[61,64,65].

Carbohydrate metabolism is linked to IIS (e.g., refs.[66,67]) and our results for the QCM point to a link with this pathway (Fig. 3 and Supplementary Fig. 1). A gene classified as an *insulin-like*

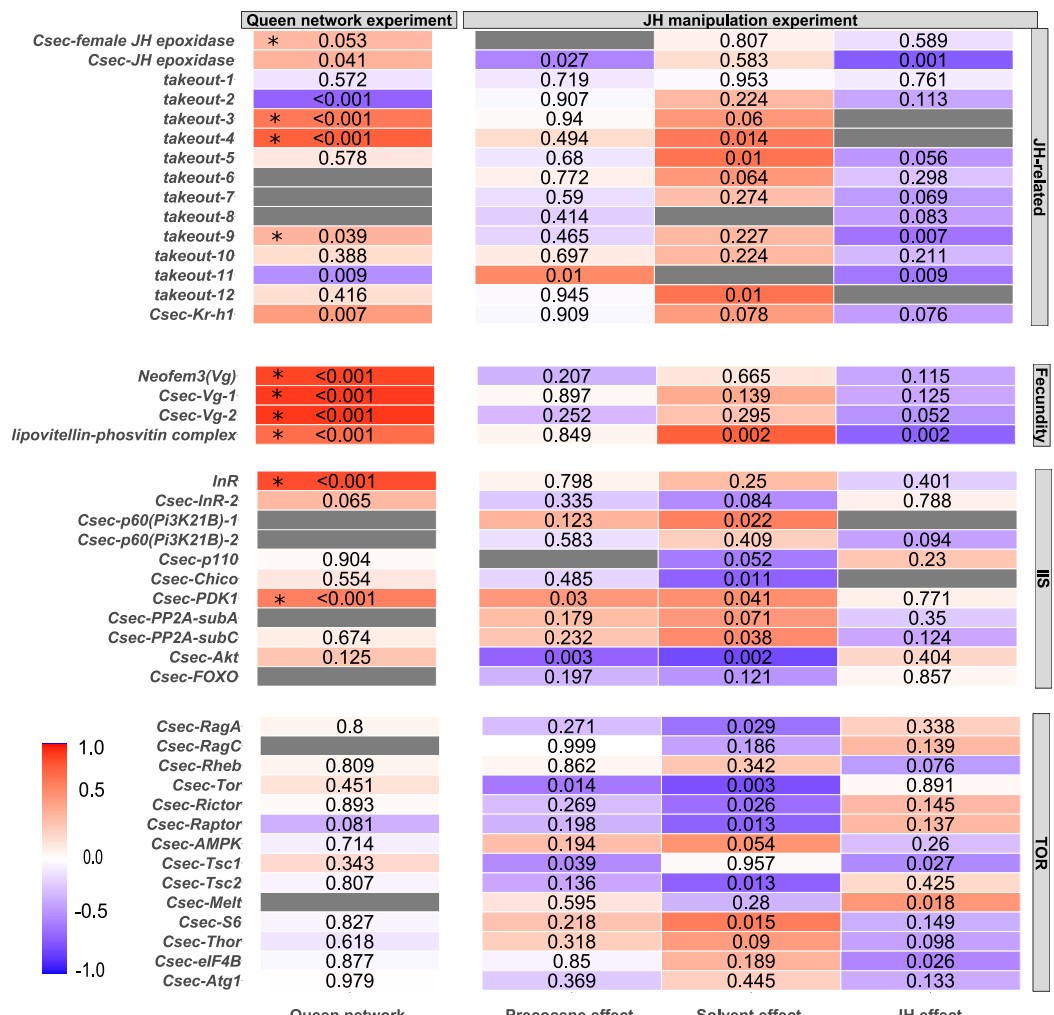

**Fig. 3 Heatmap for the WGCNA results related with JH, fecundity, IIS, and TOR.** Shown are the results of the co-expression analyses for the queen network experiment (column 1: queen network from queen–worker comparisons; $n_{worker} = 14$; $n_{queen} = 14$) and the JH-manipulation experiment with comparisons between precocene–control (column 2: precocene effect), the solvent–control (column 3: solvent effect), and the precocene–solvent (column 4: JH effect) ($n_{precocene} = 12$, $n_{solvent} = 7$, $n_{control} = 6$). Genes that are related to JH, fecundity, and nutrient-sensing pathways are listed. Each row represents a gene, with genes being sorted by pathways/functions. The color indicates how strongly a gene was associated with a trait (i.e., being a queen, precocene treatment, solvent treatment, JH effect, respectively). The strength of the association varies from −1 (i.e., blue, the strongest negative association) to 1 (i.e., red, the strongest positive association). Values close to white indicate no association. The value in each cell is the P value of the association. Gray indicates that this gene was not part of a co-expression module (i.e., it was not co-expressed) in the respective analysis. Stars * in the heatmap indicate that the gene was present in the QCM. Data are generated from Supplementary Data 3.

_growth factor 1-receptor_ (_InR_; _Csec_G15826_), an "entry-point" of the IIS pathway, as well as the _Csec-3-phophoinositide-dependent protein kinase 1 gene_ (_PDK1_), were both found in this module and were significantly associated with being a queen (Fig. 3 and Supplementary Fig. 1). In addition, further genes of the IIS pathway were associated with queens from other modules, among them _Csec-Akt_, _Csec-chico_, _Csec-p110_, and an Csec ILP (_Csec_G03496_), which is orthologous to dilp1,2,3,5, from _D. melanogaster_ (Supplementary Fig. 1). In line with former results[68], this suggests that the IIS pathway is upregulated in _C. secundus_ queens compared to workers. This accords with results for queens of social Hymenoptera (reviewed by refs. [27,31,32]).

Finally, there was a strong link to the production of CHCs in the QCM, not only via the hub gene, _Csec-Cytb5-r-a_ (Fig. 4). _Cytb5-r_ genes are involved in CHC synthesis as they are, for instance, required when desaturases introduce double bonds. Besides the cockroach-specific paralog _Csec-Cytb5-r-a_[53], the common cockroach ortholog _Csec-Cytb5-r-b_ was also co-expressed, together with two other genes putatively involved in

CHC production: a fatty acyl-CoA reductase (_Csec_G08169_) and a gene classified as the long-chain fatty-acid CoA ligase _Csec-acsbg2_0_ from the bubblegum family. Acyl-CoA synthases are involved in the synthesis of very-long-chain fatty acyl-CoAs[69]. In addition, two desaturase genes (from other queen modules) were identified to be upregulated in queens, a desaturase gene of type C, _Csec-Desat-C_ (queen module pink), and a A2-desaturase gene, _Csec-Desat-A2-a_ (queen module blue2).

**JH-manipulation experiment.** We pharmacologically reduced JH in queens by topical application of precocene and compared gene expression relative to untreated control queens ("precocene effect") and relative to solvent-treated queens ("JH effect"). In addition, we compared control queens to solvent-treated queens ("solvent effect"). Including the solvent treatment is important as it allows to separate treatment/handling artifacts from the true JH reduction effect. Acetone can have a toxic effect at the cellular level[48]. The solvent treatment is identical to the precocene

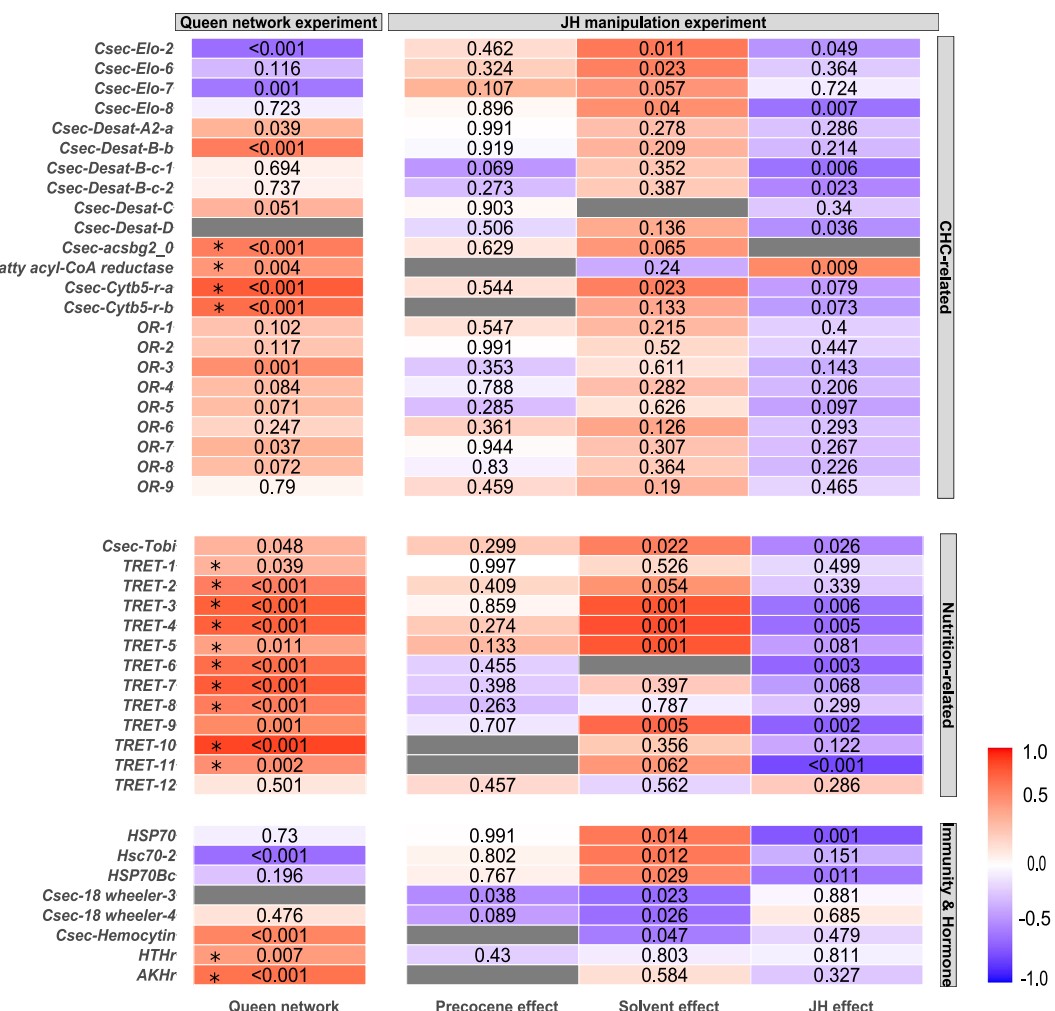

**Fig. 4 Heatmap for the WGCNA results related with CHCs, nutrition, immunity and hormones.** Shown are the results of the co-expression analyses for the queen network experiment (column 1: queen network from queen–worker comparisons; $n_{worker} = 14$; $n_{queen} = 14$) and the JH-manipulation experiment with comparisons between precocene–control (column 2: precocene effect), the solvent–control (column 3: solvent effect), and the precocene–solvent (column 4: JH effect) ($n_{precocene} = 12$, $n_{solvent} = 7$, $n_{control} = 6$). Genes that are related to cuticular hydrocarbons (CHC) biosynthesis, nutrition, immunity, and hormonal regulation are given. For more information, see Fig. 3.

treatment except that it lacks the active JH-reducing compound, precocene, which was solved in the same solvent. Using a total of 25 transcriptomes (control: $N = 6$, precocene treatment: $N = 12$, solvent: $N = 7$), we identified DEGs and gene co-expression modules (precocene versus control: "precocene modules," solvent versus control: "solvent modules," precocene versus solvent: "JH-effect modules") to test how co-expression among genes changes with JH reduction. To improve readability, we also provided functional names for the modules associated to the QCM, aiming to characterize them. We manually inferred these functions based on a number of striking genes with apparently linked functions. Note, formal enrichment analyses for the functions we were interested in are not possible, as these functions are not yet described (e.g., CHC production) or not well-represented (e.g., JH biosynthesis and signaling) in enrichment approaches, such as GO analyses. Yet, we also performed KEGG and GO enrichment analyses of each module (see Supplementary Data 5 and 6).

To obtain the precocene effect, we compared control queens with precocene-treated queens. This revealed 286 DEGs, of which 93 DEGs were upregulated and 193 DEGs were downregulated in precocene compared to control queens (Supplementary Data 2B). Applying WGCNA to identify modules of co-expressed genes resulted in 224 precocene modules. Sixteen modules had a

significantly negative association with precocene treatment (downregulated in precocene queens), while ten modules had a significant positive association with precocene treatment (upregulated in precocene queens) (Supplementary Data 3B).

Identifying modules with genes from the QCM revealed several functional modules (Figs. 3 and 4). The precocene module red can be characterized as fecundity module. It comprised seven *takeout* genes, all three *Vg* genes, the *lipovitellin–phosvitin* gene, some of the *Tret* genes, and the long-chain fatty-acid CoA ligase *Csec-acsbg2_0* that is potentially linked to CHC biosynthesis. The precocene module orange comprised genes that link JH titers with CHC synthesis. These included the early JH-response gene *Csec-Kr-h1*, one *takeout* gene, and two CHC-related elongase genes (*Csec-Elo-6*, *Csec-Elo-8*). At least one other module was associated with chemical communication, the precocene module lightsteelblue1. It included the hub gene of the QCM *Csec-Cytb5-r-a* as well as nine odorant receptor genes, two *takeout* genes, and the desaturase *Csec-Desat-B-b*. None of these modules or genes were significantly associated with precocene treatment. In addition, some genes from the IIS and TOR pathways (e.g., down in precocene: *Csec-Akt*, *Csec-Tor*; up in precocene: *Csec-PDK1*) were significantly affected by precocene treatment compared to control (Fig. 3). They were spread across many modules (Supplementary Data 3B).

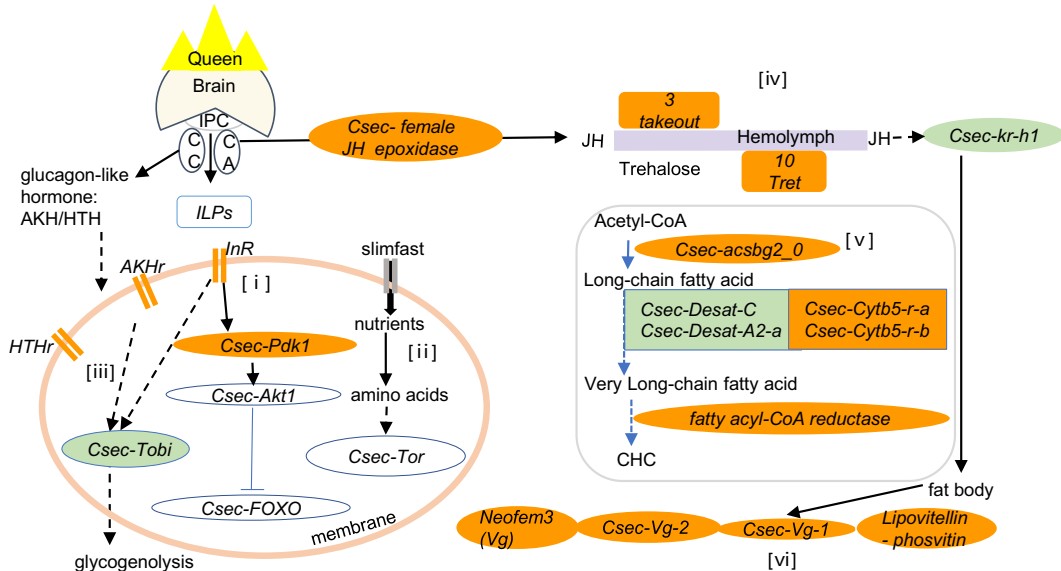

**Fig. 5 Schematic representation of the queen central module (QCM).** Shown are genes, which we found co-expressed in the QCM (orange color) and in other queen modules (greenish color) across the nutrient-sensing pathways (i) IIS and (ii) TOR, (iii) neuro-hormonal regulation of metabolic processes, (iv) JH signaling (including trehalose transport) (upper part), (v) CHC production, and (vi) vitellogenesis. The left part of the figure depicts gene regulatory pathways related to neuroendocrine and IIS/TOR signaling while the right parts concentrate on fecundity- and CHC-related processes. Both parts are connected by transport processes via the hemolymph. The general gene and pathway relationships are based on studies of *D. melanogaster*, mosquitos, nonsocial cockroaches and social insects (for details, see main text). Circles/double bars represent genes, squares represent gene 'clusters', arrows indicate activation, and stop bars denote repression. Solid versus dashed lines represent direct versus supposed indirect interactions requiring further investigation. IPC insulin producing cells, CC corpus cardiacum, CA corpora allata. For further information and gene names, see text.

Overall, we identified several functional modules related to JH that represent different hallmarks of queenness, but, surprisingly, none was significant. By contrast, several significantly affected IIS and TOR genes were distributed across many modules.

To disentangle a JH reduction effect from a handling/treatment effect, we did the solvent experiment, which was identical to the precocene treatment, except that it lacked the JH-reducing compound, precocene.

Comparing untreated control queens with solvent-treated queens revealed 1714 DEGs, of which 722 DEGs were upregulated and 992 DEGs were downregulated in solvent-treated queens (Supplementary Data 2C). Solvent treatment resulted in 219 solvent modules of co-expressed genes. Forty-six modules were significantly negative correlated solvent treatment and 27 modules were significantly positive correlated with solvent treatment (Supplementary Data 3C).

Exploring the solvent modules for genes from the QCM neither revealed modules functionally related to JH nor modules containing many QCM genes (Supplementary Data 3C). "Queen genes" were distributed across many modules. The module containing the most QCM genes was the solvent module royalblue with three *takeout* genes, several *Tret* genes, *Csec-Cytb5-r-a*, the lipovitellin–phosvitin-related gene, and *Csec-Kr-h1* (Figs. 3 and 4). However, in contrast to the precocene treatment, this module and almost all the above-mentioned genes were significantly upregulated with solvent treatment, whereas they were downregulated in precocene compared to control queens (Figs. 3 and 4). Another module with more than two QCM genes was solvent module lightsteelblue1 (Supplementary Data 3C). Here also, the modules and genes tended to be expressed more strongly in solvent-treated queens than control queens.

As in the precocene treatment, many important genes from the IIS and TOR pathway were affected by the solvent treatment, these genes were dispersed across many modules (Supplementary Data 3C). For instance, *Csec-Akt* and *Csec-Tor* were both

significantly downregulated in solvent-treated queens, as in the precocene treatment (Fig. 3). In addition, solvent modules seemed to be characterized by stress and immune response genes (e.g., several heat-shock proteins, HSP70, were positively associated with solvent treatment; Fig. 4). Overall, these results indicate an upregulation of stress-related genes and some IIS and TOR genes as well as a few queen genes. These genes were not co-expressed in modules related to hallmarks of queenness. This suggests that the results from the precocene treatment were strongly affected by treatment/handling artifacts and that these artifacts have to be disentangled from the true JH reduction effect. To do this, we compared solvent treatment with precocene treatment.

Comparing precocene-treated with solvent-treated queens revealed 189 DEGs, of which 72 DEGs were upregulated and 117 DEGs were downregulated in precocene-treated queens (Supplementary Data 2D). Comparing precocene-treated with solvent-treated queens resulted in 216 JH-effect modules of co-expressed genes. Twenty-one modules were significantly down-regulated and 14 modules were upregulated in precocene-treated queens (Supplementary Data 3D).

The separation of handling artifact from a specific JH reduction effect identified four JH-effect modules of co-expressed genes related to the QCM (Figs. 3 and 4) (Supplementary Data 3D). Two modules were fecundity-related: JH-effect module orangered1 (containing all three *Vg* genes, two *Tret* genes, and two *takeout* genes) and JH-effect module slateblue (containing the *Csec-female JH epoxidase*, one lipovitellin–phosvitin gene, and three *Tret* genes). The other two modules were related to chemical communication. The module JH-effect skyblue3 linked JH with CHC communication, it comprised the JH early-response gene *Csec-Kr-h1* and three *takeout* genes that encode JHBPs as well as two desaturase genes and an elongase. The fourth module, JH-effect darkmagenta, complemented the chemical communication signature with two desaturase genes, one gene encoding an elongase and a *takeout* gene. All these JH-effect modules were

significantly downregulated in precocene compared to solvent queens, as were almost all listed genes (Figs. 3 and 4).

Not a single core gene from the IIS and TOR pathways was affected by the JH effect though some downstream elements of TOR such as *Csec-eIF4B* were downregulated. These genes were not located within the QCM-related JH-effect modules (Supplementary Data 3D).

These results demonstrate that precocene treatment specifically affected the co-expression modules functionally related to fecundity (vitellogenesis) and chemical communication, which were both downregulated. Strikingly, no core IIS or TOR genes were affected. This implies that a downregulation of JH via precocene does not affect these nutrient-sensing modules but only the JH-fecundity and JH-CHC axes.

**The queen central module (QCM).** Our study revealed a single module of co-expressed genes that encompasses all hallmarks of a social insect queen at the molecular level: the QCM (Fig. 5). It contained core genes from the central IIS-JH-Vg/YP circuit that regulates life-history trade-offs in insects (ref. 8 and references therein). Furthermore, it extends to metabolic genes (such as *Tret* genes) and genes potentially involved in neuronal regulation of metabolism (such as *AKH*-related genes) and chemical communication. Thus, the QCM emerges as a core gene module underlying life-history traits and the queen phenotype. We could retrieve this module despite some variation, e.g., in the age of queens (see "Methods"), which can create noise. This supports the robustness of our results.

**Why did no other study identify a QCM?** So far, no other social insect study could identify a single module of co-expressed genes characterizing the queen phenotype. There are several potential reasons: First, QCMs may not exist in social Hymenoptera, which are the only social insects that have been studied using a network approach. In termites, QCMs would not have been picked up even if common due to paucity of appropriate studies. Second, the QCM might be specific to *C. secundus* and not the rule for termites. This needs further studies. Third, its detection depends on the tissue being analyzed. We chose heads with prothorax because this guaranteed that we obtained gene expression signatures of the brain (incl. antennae) and the JH producing corpora allata, as well as hemolymph and fat body. The head and prothorax are thus arguably the most suitable tissues to obtain an expression signature of the IIS-JH-Vg/YP circuit. In contrast, most other social insect studies either used whole bodies (with and without gut) or concentrated on specific tissues, such as the fat body or brain, so that part of this circuit's signature is naturally missing. Our approach, however, also meant that we missed a signal from the ovaries and may be a stronger fat body signal. This needs to be addressed in future studies. We think, however, that the head–prothorax tissue is well suited to investigate the IIS-JH-Vg/YP circuit and our results support this.

**Rewiring along the IIS-JH-Vg/YP circuit?** Our results imply that the principal connections along this endocrine circuit are not rewired in this termite species (Fig. 1) and thus, cannot explain the overcoming of the fecundity/longevity trade-off. This contrasts to what has been suggested for social Hymenoptera[8,9] and shown for the honeybee (e.g., [9,70]. First, like in most reproducing females of nonsocial insects (ref. [11] and references therein[8]), the QCM showed that reproducing *C. secundus* queens are characterized by an upregulation of the IIS, JH, and Vg components of the circuit when compared to workers. High JH titers of reproducing termite queens have also been shown in previous studies (*C. secundus*:[44]; for termites in

general:[71,72] and references therein). Second, reduced JH in queens did not appear to have feedforward effects on IIS, after controlling for solvent effects. We neither found a downregulation of any IIS genes nor a module with co-expressed IIS genes. By contrast, such a feedback loop between JH and IIS may exist during soldier differentiation in the termite *Hodotermopsis sjostedti* as gene expression analyses of selected genes using qPCR imply[73]. Third, manipulation of the central element along the IIS-JH-Vg/YP circuit, the endocrine regulator JH, demonstrated strong effects on Vg. Reducing JH apparently led to lessened vitellogenesis. This is reflected in the downregulation of two fecundity-related modules (JH-effect modules orangered1 and slateblue), which both contained co-downregulated JH-related genes (e.g., *takeout* genes) as well as vitellogenesis-related genes (e.g., *Vgs*) (Fig. 3). All these findings accord with what is known about nonsocial insects, like *D. melanogaster* (e.g., refs. [8,10,11,40,51] and reference therein). Hence, rewiring of interactions along these major nodes of the circuit cannot explain how termite queens can overcome the fecundity/longevity trade-off. How can the remolding of the fecundity/longevity then be explained? Currently, this is speculation, yet there are several possibilities. It might be that not the major connections/nodes are rewired but parts within, especially the downstream modules. For instance, how JH regulates fecundity is not changed but interactions within the fecundity module may be altered so that an increased fecundity does not go along with faster ageing. Some support for this hypothesis comes from a recent comparative study addressing termites, bees, and ants, which shows that especially the downstream components of the TI-J-LiFe network (i.e., the LiFe part) differed between young and old workers and reproductives[74]. Alternatively, though mutually not exclusively, duplications of some genes, like the *InR* genes, might have opened options to overcome the trade-off.

**Fertility signaling.** Our study provides the first strong support for the supposed mechanistic link between pathways related to fecundity and those related to CHC biosynthesis. Such a link is required for CHCs to function as fertility indicators, as has often been hypothesized (e.g., refs. [18,19,21–23] and references therein). Other compounds may also serve as fertility indicators[16,17]. Yet for *C. secundus*, former work already pointed to CHCs[20] and this is line with the gene expression results from the current study. The QCM contained several genes related to CHC biosynthesis, including the hub gene *Csec-Cytb5-r-a*. In addition, two other queen modules were found with a strong chemical communication-related signal, one for CHC synthesis (queen module brown1) and the other for CHC perception (queen module blue2). This correlational evidence is strengthened by our experimental results. They demonstrated a causal link between a reduction in JH and a significant downregulation of a JH-CHC-related module (JH-effect module skyblue3) that co-expressed genes related to CHC biosynthesis (two desaturases and an elongase) as well as JH signaling (including the early JH-response gene *Csec-Kr-h1*). In addition, a CHC module (JH-effect module darkmagenta) was significantly affected. Strikingly, reproducing *C. secundus* queens are characterized by long-chain CHCs, including alkenes, that qualitatively distinguish them from workers[20,75]. Thus, the identified genes are promising candidates of fertility indicators as similar genes in other species[76–78] encode enzymes typically involved in the introduction of double bonds (desaturases) and the elongation (elongases) of CHCs.

Communication is an essential characteristic of social life. The striking links to social communication that we revealed offer new avenues to understand the remodeling of the fecundity/longevity trade-off in social insects.

## Methods

**Species collection and colony maintenance.** *Cryptotermes secundus* colonies were collected from dead *Ceriops tagal* mangrove trees near Palmerston-Channel Island (Darwin Harbor, Northern Territory, Australia; 12°30′S 131°00′E). Colony collection, transportation, and housing were performed as previously described in refs. [79,80]. In short, colonies were kept in *Pinus radiata* wood blocks in climate rooms in Germany, providing 28 °C, 70% relative humidity, and a 12-h day/night cycle. Under these conditions, colonies develop as in the field[79,80].

We set up over 80 groups of at least 30 workers from which queens (neotenic replacement reproductives) develop after about 2 weeks (for details see refs. [43,81]). From these, 39 groups which had developed well (i.e., they had a pair of fertile reproducing neotenics) were used for the study (14 for the queen network + 25 for the JH-manipulation experiment). The different queens and workers used in this study all came from different (stock) colonies. Thus, the data represent independent biological replicates. For our experiments, we used queens with a maximum age of 2 years that were laying eggs. For sample details see Supplementary Data 1.

**Experiment 1. Uncovering the molecular queen network (=queen network experiment).** To identify the genes positively associated with queens, we compared workers and queens. We dissected the heads and prothorax (to ensure inclusion of the corpora allata) of reproducing queens, and one arbitrarily selected worker of the same colony, from a total of 14 different colonies to generate 28 head/prothorax transcriptomes (2 castes × 14 colonies) (Fig. 2b).

**Experiment 2. JH-manipulation experiment.** We pharmacologically reduced the main insect gonadotropic hormone, JH, in queens to test how JH reduction affects network connections along the IIS-JH-Vg circuit. We had two treatments, which we compared to six untreated queens from different colonies: a precocene and a solvent treatment (Fig. 2c).

For the precocene treatment, 12 queens from different colonies were treated with 1.5 μl of a 0.1% acetone solution of precocene I (99% Sigma-Aldrich) topically applied to the head (0.5 μl) and thorax (1 μl) to reduce their JH titers. Precocenes are 2,2-dimethyl chromene derivates originally found in plants that interfere with the JH biosynthesis in the corpora allata of insects[82]. The applied concentration generally reduces hemolymph JH titers of *C. secundus* queens to undetectable levels, while not being detrimental to queens' overall health. Precocene treatment also resulted in behavioral changes of the workers indicative of queenless colonies and in altered changed CHC profiles as expected for less-fertile queens. After 14 h, we dissected the head plus thorax of queens for transcriptome generation.

For the solvent treatment, which controls for handling and solvent-induced effects, we performed the same experiment as above, but applied only the solvent pure acetone (0.5 μl on head and 1 μl on thorax) to seven queens from different colonies.

**Transcriptome: RNA extraction and sequencing.** Tissues were preserved in RNAlater (Qiagen) first at 4 °C for 24 h and then at −20 °C until extraction. RNA was extracted from head plus thorax via single step isolation, as follows. First, thawed tissue of an individual was mixed with peqGOLD TriFast™ (Peqlab) and homogenized for 2–3 min. We then added chloroform, separated the aqueous phase, added nuclease-free glycogen (5 mg/ml) and isopropanol (Ambion) to precipitate total RNA. 75% ethanol and nuclease-free water were used to wash and dissolve the pellet, respectively. After washing the RNA pellet with 75% ethanol and vortexing, the sample was centrifuged for 5 min at 4 °C, 8500 rpm. This step was repeated three times. After dissolving the pellet in nuclease-free water, samples were kept for at least 2 h at 4 °C. DNA digestion was performed using the DNase I Amplification Grade Kit (Sigma-Aldrich). The concentration and quality of the isolated RNA were checked with an Agilent Bioanalyzer (Agilent RNA 6000 Nano Kit) and sent to BGI, Hong Kong and then to Shenzhen (PR China) on dry ice. After quality check, library preparation was done by BGI using the TruSeq RNA Library Prep Kit v2 (Illumina). Amplified libraries were sequenced on an Illumina HiSeq4000 platform (100-bp paired-end reads), generating ~4 Gigabases of raw data for each sample. After sequencing, index sequences from the machine reads were demultiplexed (sorted and removed) by a proprietary BGI-inhouse tool.

**Processing of RNASeq raw reads.** Raw reads were trimmed using fastp version 0.19.6[83] and Trimmomatic (version 0.38) (for details, see Supplementary Methods). Trimmed results were checked using MultiQC-1.7[84] and quality of reads was assessed with FastQC[85] (version 0.11.5). Trimmed sequences were mapped to the draft *C. secundus* genome GCA_002891405.2_Csec_1.0[53] using HISAT2 (version 2.1.0). SAM files generated by HISAT2 were first converted to BAM files and then sorted by name using SAMtools (version 1.9). The sorted BAM files were put into HTSeq[86] to generate gene count tables using the following settings: -f bam, -i ID, -t gene, -m union, -r name, --stranded=no.

**Differential gene expression analysis.** DEGs between queens and workers, and between control and treatment queens, were determined using the generalized negative binomial model implemented in DESeq2[87](version 1.18.1) in R (R Core Team, version 3.5.2). *P* values were calculated using the Wald test and corrected for

multiple testing using the false discovery rate (FDR) approach[88]. Genes were listed as DEGs if their corrected *P* value was smaller than or equal to 0.05. To check for caste and treatment effects on gene expression profiles, gene count data were normalized using the varianceStabilizingTransformation function from DESeq2. The top 500 genes with the highest row variance were used to perform principal component analysis (PCA) using the plotPCA function from the DESeq2 package (Supplementary Fig. 2).

**Network analysis.** To identify networks of co-expressed genes (modules), we used weighted gene co-expression analysis (WGCNA)[89,90]. These analyses were done separately for each experimental comparison, so we had four comparisons in total: queen network (queens versus workers); precocene effect (precocene-treated queens versus control queens); solvent effect (solvent-treated queens versus control queens); and low JH effect (precocene-treated versus solvent-treated queens).

**Preprocessing data for WGCNA: outlier detection and gene filtering.** As recommended by ref. [91], we performed several preprocessing analyses to guarantee high data quality. We did hierarchical clustering to detect outliers using the function hclust in the R package flashClust (version 1.01-2). Samples clustering together were in the same treatment group, which suggests that there were no obvious outlier samples that need to be removed. Samples and genes with more than 50% missing values and zero variance were filtered iteratively using the goodSamplesGenes function with default settings in the WGCNA package. No samples had to be excluded but 2103, 1840, 1927, and 1931 genes were removed due to poor quality in the queen network, precocene-effect, solvent-effect, and low JH-effect co-expression analyses, respectively. We used all genes, not only DEGs, as a restriction to DEGs might bias the co-expression network and invalidates the scale-free topology assumption of WGCNA.

**WGCNA.** WGCNA was conducted using the package WGCNA[90] (version 1.68) for all expressed genes that passed quality filtering. Gene counts were first normalized using the varianceStabilizingTransformation function from DESeq2[87] (version 1.18.1). Variance-transformed data were then used to construct a signed adjacency matrix with different soft-threshold powers that fit the scale-free topology assumption best, with a relatively high (i.e., more than 10) mean connectivity. For the queen network experiment and the precocene–control analyses the best soft-threshold power was 15; for the solvent–control and the precocene–solvent analyses it was 11. Adjacency-based dissimilarity (an alternative of dissimilarity based on the topological overlap matrix) was used for an average linkage hierarchical clustering analysis. This approach allowed us to identify modules containing genes from different biological pathways. The resulting dendrogram was then used to detect modules (i.e., clusters of co-expressed genes that were positively correlated with each other; minimum cluster size 30) using the cutreeDynamic function from the package dynamicTreeCut[92] (version 1.63.1). To investigate module properties, the modules' eigengene (i.e., a module's first principal component) was determined with the moduleEigengenes function. Eigengenes were then used as a representative of each module to calculate the module–trait (e.g., queen–worker; precocene–control) (weighted) association and its corresponding Student asymptotic *P* value. The hub gene (i.e., the gene with the highest connectivity within a module) of each module was determined using the chooseTopHubInEachModule function. Gene–trait associations and their corresponding Student *P* values were calculated using the corAndPvalue function. *P* values that were smaller than or equal to 0.05 were considered as statistically significant.

**Genome-wide annotation.** A draft version of the *C. secundus* genome was used to obtain nucleotide and protein sequences[53]. All amino acid sequences (30,648) from the official gene set were annotated at the genome level using InterProScan v5.33-72.0[93] with the settings: -pa -goterms, others as default. InterProScan integrates 15 databases such as Pfam, Gene3D, and SMART, which allows comprehensive functional annotation. In addition, a local BLAST search was performed using BLASTP (version 2.4.0) against the fruit fly *D. melanogaster* (version 6.27), the cockroach *Blattella germanica* (version OGS1.2: https://i5k.nal.usda.gov/content/data-downloads[53]), and the two termites *Macrotermes natalensis* (official gene set version 1.2; Poulsen et al, 2014) and *Zootermopsis nevadensis* (version 2.2: http://termitegenome.org/?q=consortium_datasets[60]). We took the first (i.e., the hit with lowest e value and highest bitscore) of the twenty hits and used hits with e values < 1e⁻⁵. This resulted in 21,642 hits with *D. melanogaster*, 26,238 hits with *B. germanica*, 26,806 hits with *M. natalensis*, and 25,154 hits with *Z. nevadensis*. Functional annotations for *M. natalensis* were downloaded from the SI of ref. [94] and for *D. melanogaster* from UniProt (2019). We retrieved FlyBaseID information from FlyBase[95] (FB2020_01, released Feb 12, 2020) to convert *D. melanogaster* protein IDs into gene IDs, thus, matching them with the functional annotation downloaded from UniProt. Blast results of *B. germanica* and *Z. nevadensis* lacked functional annotations. Hence, we provided only a hit-table in the supplement (Supplementary Data 7).

**Identification of the modules related to hallmarks of queenness.** In order to identify modules that could functionally be related to hallmarks of queenness, we manually explored all available genome annotations. Whenever possible, genes

were classified into functional categories such as vitellogenesis, JH-related, related to chemical communication, related to immunity, P450 genes, as well as genes from the IIS and TOR pathways. As guideline, we used the TI-J-LiFe gene list (Supplementary Data 4), a list of genes from the TI-J-LiFe network that combines all major pathways underlying aging and life-history trade-offs. This list was originally generated for *D. melanogaster*[10,40,49–52]. All genes of this list, as well as all genes potentially involved in chemical communication, have been well characterized for *C. secundus*, including the construction of gene trees[53]. They are presented in the text as *Csec-XXX* genes with XXX standing for the gene name. The few genes for which we only had annotation results, are presented by their Gene ID *Csec_-GYYYYYY* with YYYYY standing for five numbers.

**Gene ontology (GO) enrichment analysis**. To gain insights into the biological processes that characterize a module, we performed GO enrichment analysis for all genes of each WGCNA module using the "weight01" algorithm with "fisher" test in the R package TopGO[96] (version 2.32.0) (Supplementary Data 6). GO annotations were extracted from InterProScan results and all the expressed genes with GO annotations were used as background for enrichment analyses.

**KEGG pathway analysis**. To gain insights into the molecular pathways that characterize a module, we performed KEGG enrichment analysis for all genes of each WGCNA module. KEGG pathway identifiers were obtained from the result of InterProScan and pathway information was retrieved from KEGG official website (https://www.genome.jp/kegg/pathway.html#energy). Enrichment analysis was performed using the function *enricher* from the R package clusterProfiler[97] (version 3.8.1). It tests for over-representation of KEGG pathways with the hypergeometric test and adjusts *P* values using the FDR approach[88]. All genes with KEGG annotations in the whole genome (771 genes) were used as background for enrichment analyses (Supplementary Data 5).

**Reporting summary**. Further information on research design is available in the Nature Research Reporting Summary linked to this article.

## Data availability
The raw sequence data used in this publication are deposited at the National Center for Biotechnology Information (NCBI) under the Umbrella Project "SoLong" (BioProject IDs PRJNA654896, PRJNA683654, PRJNA685294). Source data underlying Figs. 3 and 4 are presented in Supplementary Data 3. Additional data are available from the corresponding author upon reasonable request.

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

## Acknowledgements

We thank Florentine Schaub for assistance in the field and wet lab, Karen Meusemann for help with bioinformatics analyses, Daniela Schnaiter for handling and maintenance of the termite colonies, Jonathan Henshaw for revising our English, and Veronika Rau for discussions about analyses. We further thank the editors and three anonymous reviewers for providing helpful comments that improved the manuscript. We acknowledge support by the Baden-Württemberg High Performance Computing facilities. Charles Darwin University (Australia), and especially S. Garnett and the Horticulture and Aquaculture team, provided logistic support to collect *C. secundus*. The Parks and Wildlife Commission, Northern Territory, and the Department of the Environment, Water, Heritage and the Arts, Australia,

gave permission to collect (Permit number 59044) and export (PWS2016-001559) the termites. The study was conducted in accordance with the Nagoya protocol. This research was supported by the Deutsche Forschungsgemeinschaft with two grants to J.K. (DFG; KO1895/23-1; KO1895/25-1), one of these within the Research Unit FOR2281, and a stipend to J.W. by the Carl-Zeiss-Stiftung (0563-2.8/685/2).

## Author contributions

J.K. designed the study, collected and identified the termite samples, S.L. and J.W. performed all experiments and did the transcriptome analyses, J.K. supervised the study, and all authors wrote the paper.

## Funding

## Competing interests

The authors declare no competing interests.
