## [Peer Review File · Communications Biology]

Reviewers' comments:

Reviewer #1 (Remarks to the Author):

The manuscript untitled "Live fast, die old: a gene network underlying long lifespan and high fecundity" intends to investigate if re-wiring along the nutrient-sensing / endocrine / fecundity axis can account for the reversal of the fecundity / longevity trade-off in social insect queens by using the termite *Cryptotermes secundus*. They have identified a single module of co-expressed genes that characterized queens in this species and that encompassed genes from all essential pathways known to be involved in life-history regulation in solitary model organisms. The data seem sound and they are novel. They will be of interest to others in the termite community as well as the social insects community and all scientists interested in fecundity and longevity. This paper will influence thinking in the termite field by providing a new way, different from what it is considered by now. Researchers may reproduce the work provided here, given the level of detail provided. The manuscript is well written and in the scope of the journal.

I understand the choice for the head and the prothorax (line 328) but I believe there is not much fat body inside and it does not necessarily reflect fecundity. The abdomen would have been clearly the best localization, especially since it is next to the ovaries.

I was surprised not to see any behavioral controls prior to dissections, to be sure that the JH manipulation experiment worked out (especially since only 14 hours were left between precocene application and dissections). The typical queen behavior has already been characterized in this termite species by Korb's team. Moreover, another subset of queens could have been used for CHC profiles after precocene application by using the same protocol.

It is written that termite queens can live for decades (lines 26-27). However, it is apparently not the case for all termite queens (e.g. Keller et al. 1998). According to Monroy Kuhn et al. (2019), the median maximum lifespan of *Cryptotermes secundus* is only 6-7 years so I am not sure that the choice for this termite species is the best to reply to this question of trade-off between fecundity and longevity, even if workers do not leave long. The authors even say that their results might be specific to *C. secundus* and not the rule for termites (line 327).

What about kings? They do not lay eggs of course but they still reproduce with the queen and they live as long as the queens.

In conclusion, even though I leveled some criticisms, I believe that this study deserves to be published in *Communications Biology* after modification. Here is the list of some minor modifications and questions :

-L.27 : « live » instead of « life ».

-L.27 : It will be good to give the mean of the lifespan of solitary insects instead of what the authors suspect.

-L. 35 : TOR is mentioned in the text which refers to figure 1 but TOR does not appear in figure 1.

-L. 57 : "which" instead of "who".

-L. 61 : Which family?

-L. 63-66 : And what about Hanus et al. 2009 Beyond cuticular hydrocarbons: evidence of proteinaceous secretion specific to termite kings and queens. *Proc. R. Soc. B* ? You may mention this reference which gives another look.

-L. 91 : 2004).

-L. 124-125 : Well, it is typical control, I do not think it is needed to say "crucial" to describe controls.

-L. 206 : 2018).

-L. 363 : It has been often hypothesized, indeed, but other hypotheses which do not go in the same direction have been done as well. I believe they should be mentioned in this paragraph.

-L. 393 : How many groups?

-L. 394 : How long after the queens differentiate did the experiments start? This information seems important to me to be sure that all the queens were completely differentiated (morphologically and physiologically) even though they were laying eggs.

L. 779 : "activation, respectively".

Figure 2 : The part about Queen network experiment does not bring anything in this figure and the few very details provided could be given in the text itself. Actually, it brings more confusion since the JH experiment application is indicated in this figure as well and it looks like all the experiments are done one after the other by putting them all in the same figure.

Figure 2 : Two workers are on the back. Don't you have a better picture (more natural position of the termites)?

Reviewer #2 (Remarks to the Author):

General comments.

I found this study fascinating and I enjoyed very much reviewing it. The sample size was very good. I found very interesting the use of the WGCNA analysis to found modules of coregulated genes. This new type of analysis seems for me appropriate to find new gene networks associated with specific biological traits. The article is well written and the working hypothesis is clear. Still, I have several suggestions aimed to improve the clarity of some parts of the manuscript.

Specific Comments

1. Lines 16-19.

"By manipulating its endocrine component, we tested the most recent hypothesis that re-wiring along the nutrient-sensing / endocrine / fecundity axis can account for the reversal of the fecundity / longevity trade-off in social insect queens. Our data do not support this hypothesis".

R: The authors make a general conclusion for social insect queens based on what they found in a particular type of social insects. I think that they can either conclude that such a hypothesis is not supported in termites or that termites are different from what happens in other social insects (e.g., honey bees).

2. Line 125-128.

"Head-prothorax transcriptomes appear well suited for our purpose as they likely encompass components from neuro-endocrine signaling (e.g. linked the JH biosynthesis by the corpora allata), chemical perception and production (antenna and prothorax, respectively), as well as fat body related vitellogenesis".

R: This study lacks the ovaries, which also are a presumably key element to explain the queen's fertility. While I think that there are advantages and disadvantages of using the whole body, including the ovaries, versus only the head-prothorax; I think that it could be good someplace in the manuscript, to acknowledge that this study did not take into consideration the ovaries and that this may provide additional insights into the understanding of the trade-off between fertility and longevity.

3. Lines 132-133.

"Analyses of 28 transcriptomes (14 queens and 14 workers)" ^[L1]_[SEP]

R: This number of transcriptomes looks good. However, there is a lack of information on more specific details of the samples to confirm that they can be considered true replicates. For example, are they of similar age and reproductive condition? I understand that the samples for queen/worker comparison (Data set 1 and Data set 2 A) were collected in the field (data set 1) and the age of the termites is likely unknown, but I could appreciate if the authors could clarify this. On the other hand, in the hormonal manipulation experiments, where the termites were kept in the lab, which were the nutritional conditions used? Can the authors, for example, discard that the collected termites were not experienced nutritional stress?

4. Lines 174-185. Vg genes

I am extremely curious in knowing more about the function of the different Vg genes in termites. In ants, Vg duplicated and experienced different molecular evolution to acquired caste and behavioral specific functions (e.g., Wurm et al, 2011; Corona et., 2013, Kohlmeler, 2018). What happens in termites? I understand that all the Vg are upregulated in *Cryptotermes* queens. Any clue if they have different functions? Since there is only a true Vg in bees and wasps, termites and ants are the other social insect groups with several Vg genes. I think that it could be very interesting investigate if Vg in termites also underwent subfunctionalization.

5. Line 164. JH signal

I agree that high JH epoxidase expression levels suggest increased JH titers, as also have been shown in honey bees and ants. Also, I agree that higher expression of other genes related with JH signal, such as Kruppel is consistent with this view. Please clarify if they are two genes (or isoforms) encoding possible JH epoxidases in *Cryptotermes* as suggested in the figure 3.

6. Lines 199-206. IIS.

R: I think that here it needs it a little more information on the key elements of the IIS in termites. First, how many insulin receptors are encoded in the *Cryptotermes* genome? There are indeed two, as suggested in figure 3? Second, how many insulin-like peptides? What happens with the expression of these genes? Unfortunately, the authors do not report caste-specific differences in the expression of these genes that are key to understand if the IIS signal is up-regulated in queens compared with workers as it is suggested. If these genes were not detected (they are normally expressed at very low levels) please clarify. In conclusion, while I agree that higher InR and PDK1 expression in queens suggest a higher insulin signal in queens, I still found the evidence not quite strong without reporting the expression of insulin genes.

7. Hormonal manipulations.

In the first comparison, the authors did not find significant differences in the expression of key genes between the control queens and the precocene treatment. However, when the effect of the solvent was uncoupled, significant differences were apparent in the direction contrary to the honey bee model. Then, If I am right, the solvent masked the effect of precocene. I am not sure if the authors discussed enough why this happened. I think this may be relevant to understand better the results. For example, does JH had a JH-like effect, maybe related to a stress response?

Reviewer #3 (Remarks to the Author):

This study seeks to identify the mechanisms underlying a reversal of life history tradeoffs that has occurred in queens of social insects. The authors use transcriptomes in termite queens to (1) characterize expression modules related to queen function and (2) experimentally test how juvenile hormone influences other components of the molecular pathways associated with life history tradeoffs. They find a module of coexpressed genes that has elements of all the important queen functions, including fecundity, juvenile hormone synthesis, chemical communication, and immunity. They do not, however, find evidence that the set of pathways regulating life history tradeoffs has been “re-wired” in termites. I feel this study addresses an important question with an exciting hypothesis. I appreciated the experimental manipulation included in the study. The methods were mostly rigorous and the paper was well written. In general, I found this to be a compelling study. I do have a couple of outstanding questions, which if addressed, may improve the clarity and impact of the paper.

First, the experimental part of the study yielded largely negative results. The hypothesis that the IIS-JH-Vg pathway has been rewired is not supported by the precocene treatments in termites. However, this leaves open the question of how termite queens have had a reversal of life history trade-offs and this is never really explained. For example, the Introduction states that high levels of JH in solitary insects increase fecundity, but also have several life shortening effects. Yet this study reveals that the positive relationship between JH and vg/fecundity seems to be maintained in termites and that queens seem to have an activated JH pathway. So how then is it possible that queens are so long-lived? While this question seems to be driving the research, it does not seem to really be answered by the results. It may be interesting for the authors to speculate a bit here. It might even be interesting to include a fourth panel to Fig 1 illustrating a hypothesized scenario for what might be happening in termite queens. What are we to make of the re-wiring hypothesis?

I had some methodological questions about the WGCNA analysis of the precocene experiment. I don't understand why the precocene-treated queens compared to untreated control queens is labeled as the precocene effect. Isn't this really the precocene + solvent effect? It seems to me you would want to isolate the effect of JH by first identifying genes that are differentially expressed between precocene-treated vs control AND precocene-treated vs solvent-treated, but NOT between solvent-treated and control. Then it seems you would want to map these onto the QCM. As it is currently, it seems that WGCNA was applied separately to each treatment group and then modules containing genes from the QCM were identified. I am unclear what the rationale for this is or what it tells us. For example, it is not intuitive to me how to interpret the distribution of genes in the QCM among WGCNA modules in the solvent-treated queens.

Further, I am confused about how the WGCNA was done. In the section 'Disentangling JH reduction from handling artifacts', it sounds as though perhaps only genes that are differentially expressed between precocene-treated and solvent-treated queens were used in the WGCNA, but this seems to contradict what is stated on L136 and L475.

I also found myself questioning the choice of tissue used for transcriptomics. Some of this justification came at the very end, but it would be helpful to this reader if that justification came earlier. Fig. 4 goes so far as to hypothesize the functional pathways of the newly discovered QCM, including purported function in the abdomen. I am not clear on how the coexpression modules detected in the head/prothorax can inform us about molecular pathways operating in other tissues of the body. L176 points out that vitellogenins are produced in the fat body and carried through the hemolymph to the ovary for oogenesis. Does this include fat body in the head?

Some specific questions about methods

L393: How many colonies were used to generate the queens in this study? Were any of them related? If so, I would think this would need to be accounted for in statistical analyses.

L294: "identified four JH-effect modules of co-expressed genes related to the QCM". Does this just mean genes belonging to the QCM were found in these modules? Or was there significant overlap among the modules? Attributes to the modules are being stated as they relate to the function of just a few genes. For example, JH-effect module orangered1 is described as containing 7 genes related to fecundity. But is this a significant enrichment? How many genes are in this module? This is a recurring issue throughout this section.

L 462: It sounds like just the DEGs for each comparison is included in the WGCNA, but L136 and L475 seems to indicate all genes were used. If the latter, then what does the comparison refer to here?

L542: Where were the results of the KEGG pathway enrichment tests presented?

Responses to Reviewer comments

Reviewer #1:

1. The manuscript untitled "Live fast, die old: a gene network underlying long lifespan and high fecundity" intends to investigate if re-wiring along the nutrient-sensing / endocrine / fecundity axis can account for the reversal of the fecundity / longevity trade-off in social insect queens by using the termite *Cryptotermes secundus*. They have identified a single module of co-expressed genes that characterized queens in this species and that encompassed genes from all essential pathways known to be involved in life-history regulation in solitary model organisms. The data seem sound and they is novel. They will they be of interest to others in the termite community as well the social insects community and all scientists interested in fecundity and longevity. This paper will influence thinking in the termite field by providing a new way, different from what it is consider by now. Researchers may reproduce the work provided here, given the level of detail provided. The manuscript is well written and in the scope of the journal.

Author Reply: Thank you for your positive words and helpful comments. Your comments have helped us greatly to clarify a number of important points which – we have to admit – were insufficiently clear in our original manuscript. We have now substantially revised our manuscript in the light of your comments.

2. I understand the choice for the head and the prothorax (line 328) but I believe there is not much fat body inside and it does not reflect necessarily the fecundity. The abdomen would have been clearly the best localization, especially since it is next to the ovaries.

Author Reply: We agree that fecundity signals should be strongest in the abdomen. We decided to use head/prothorax instead for the following reasons: (i) we wanted to investigate fecundity regulation by JH to test the hypothesis about rewiring along this axis (see also JH manipulation experiment). Therefore, we needed the JH biosynthesis part and, thus, the corpora allata which is located in the head prothorax region, and (ii) we had hoped – and as it turned out, we were right - to also obtain fat body and hemolymph signals since they are included in the prothorax. Ideally, we would also have added fat body transcriptomes but (i) the quality of fat body RNA which we extracted was poor and (ii) the sample sizes for the head/prothorax transcriptomes would have been reduced by adding fat body transcriptomes due to money restraints. The latter would have undermined the robustness of especially WGCNA, which recommends at least 15 samples for the analysis (Langfelder & Horvath 2008). We recognize that this was not laid out very well and we therefore added information to the introduction (l. 109-113) and the results section (l. 320-329).

3. I was surprised not to see any behavioral controls prior dissections, to be sure that the JH manipulation experiment worked out (especially since only 14 hours were left between precocene application and dissections). The typical queen behavior has already been characterized in this termite species by Korb's team. Moreover, another subset of queens could have been used for CHC profiles after precocene application by using the same protocol.

Author Reply: We agree and we actually tested for this. After 14h the workers behaved as if they sit in queenless colonies (as expected) and also the CHC profile changed. Yet, these data

together with other data are part of another extensive manuscript. We have added a corresponding sentence (l. 412-414).

4. It is written that termite queens can live for decades (lines 26-27). However, it is apparently not the case for all termite queens (e.g. Keller et al. 1998). According to Monroy Kuhn et al. (2019), the median maximum lifespan of *Cryptotermes secundus* is only 6–7 years so I am not sure that the choice for this termite species is the best to reply to this question of trade-off between fecundity and longevity, even if workers do not leave long. The authors even say that their results might be specific to *C. secundus* and not the rule for termites (line 327).

Author Reply: We agree that *C. secundus* does not have the most long-lived reproductives in termites. Yet, please note that 6-7 years is the median maximum lifespan (also for *Macrotermes bellicosus* which has reproductives that can live for over 20 years, the median maximum lifespan is 'only' 9-10 years). *C. secundus* queens can live for up to 13 years (unpubl. data) and even 6-7 years is much longer than the lifespan of a solitary insect and it is associated with reproduction. Nevertheless, we cautiously interpreted our results as "might be specific to *C. secundus* and not the rule for termites" (l. 318-319). Whether this QCM is more universal in termites can be tested in upcoming studies.

5. What about kings? They do not lay eggs of course but they still reproduce with the queen and they live as long as the queens.

Author Reply: This is a very good point and it would be interesting to investigate it in the future! However, this question itself can be another independent study that might require more work to address since much less is known for kings than for queens.

6. In conclusion, even though I leveled some criticisms, I believe that this study deserves to be published in *Communications Biology* after modification. Here is the list of some minor modifications and questions :

-L.27 : « live » instead of « life ».

Author Reply: Thanks for pointing it out. We have corrected it (l. 19).

7. -L.27: It will be good to give the mean of the lifespan of solitary insects instead of what the authors suspect.

Author Reply: We changed it in the main text. According to Keller & Genoud (1997), the mean lifespan of solitary insects is 0.1 ± 0.2 y (l. 20).

8. -L. 35 : TOR is mentioned in the text which refers to figure 1 but TOR does not appear in figure 1.

Author Reply: This is correct. The TOR pathway is not directly included in the IIS-JH-Vg/YP circuit and is not addressed in the 're-wiring hypothesis'. We changed it in the figure legend of Figure 1 (l. 793-795).

9. -L. 57 : "which" instead of "who".

Author Reply: Thanks for pointing this out. We corrected it (l. 46).

10. -L. 61 : Which family?

Author Reply: We added that it is the Termitidae (l. 50).

11. -L. 63-66 : And what about Hanus et al. 2009 Beyond cuticular hydrocarbons: evidence of proteinaceous secretion specific to termite kings and queens. Proc. R. Soc. B ? You may mention this reference which gives another look.

Author Reply: There can of course be other compounds involved that distinguish reproductives and workers. As suggested, we added the reference (l. 52, 578-580). Yet, for *C. secundus* we have strong evidence for CHCs being important (Hoffmann et al. 2014).

12. -L. 91 : 2004).

Author Reply: Thanks. We corrected it (l. 71).

13. -L. 124-125 : Well, it is typical control, I do not think it is needed to say "crucial" to describe controls.

Author Reply: We agree that this treatment is 'typical'. We changed it to 'necessary' (l. 105) to emphasise the importance of the solvent control treatment in our study to reveal the 'true' JH effect.

14. -L. 206 : 2018).

Author Reply: Thanks. We corrected it (l. 197).

15. -L. 363 : It has been often hypothesized, indeed, but other hypotheses which do not go in the same direction have been done as well. I believe they should be mentioned in this paragraph.

Author Reply: It is true that other compounds can play a role in chemical communication as well and we have added corresponding text and references (l. 52, 364-366, 578-583). Yet, for *C. secundus* we have previously gained strong evidence for CHCs being important (Hoffmann et al. 2014) and this is in line with the QCM results we found (see also above).

16. -L. 393: How many groups?

Author Reply: We set up over 80 groups of at least 30 workers from which queens (neotenic replacement reproductives) develop after about two weeks (for details see Weil et al. 2007, Hoffmann & Korb 2011). From these, 39 groups which had developed well (i.e. they had a pair of fertile reproducing neotenic) were used for the study (14 for the queen network + 25 for the JH manipulation experiment). We added this information to the text (l. 391-394).

17. -L. 394 : How long after the queens differentiate did the experiments start? This information seems important to me to be sure that all the queens were completely differentiated (morphologically and physiologically) even though they were laying eggs.

Author Reply: All queens were fully differentiated and laid eggs. They were laying eggs up to a maximum of 2 years. Thus, they represent young queens. We added this information to the text (l. 396-397). Despite some variability in the data (which might create noise), the strong WGCNA signals we found across colonies shows the robustness of the data, regardless of conditions (l. 311-313).

18. L. 779 : "activation, respectively".

Author Reply: Thanks, we corrected this (l. 799).

19. Figure 2: The part about Queen network experiment does not bring anything in this figure and the few very details provided could be given in the text itself. Actually, it brings more confusion since the JH experiment application is indicated in this figure as well and it looks like all the experiments are done one after the other by putting them all in the same figure.

Author Reply: We would like to keep the Queen network experiment part of the figure to show readers, at first glance, what we did. To clarify that these were two independent experiments, we added: experiment 1 / experiment 2 on figure 2 with further explanations in the figure legend that experiment 1 compared queens and workers, and experiment 2 compared queens under different treatments (l. 803-806, see Figure 2 below).

20. Figure 2: Two workers are on the back. Don't you have a better picture (more natural position of the termites)?

Author Reply: Yes, we changed it (see Figure 2 below).

Reviewer #2:

1. General comments. I found this study fascinating and I enjoyed very much reviewing it. The sample size was very good. I found very interesting the use of the WGCNA analysis to find modules of coreregulated genes. This new type of analysis seems for me appropriate to find new gene networks associated with specific biological traits. The article is well written and the working hypothesis is clear. Still, I have several suggestions aimed to improve the clarity of some parts of the manuscript.

Author reply: Many thanks for your positive words and your comments that helped to improve our manuscript. Please, see below for more details.

Specific Comments

2. Lines 16-19. "By manipulating its endocrine component, we tested the most recent hypothesis that re-wiring along the nutrient-sensing / endocrine / fecundity axis can account for the reversal of the fecundity / longevity trade-off in social insect queens. Our data do not support this hypothesis".
R: The authors make a general conclusion for social insect queens based on what they found in a particular type of social insects. I think that they can either conclude that such a hypothesis is not supported in termites or that termites are different from what happens in other social insects (e.g., honey bees).

Author Reply: We agree and changed the text accordingly to 'Our data from termites do not support this hypothesis.' (l. 12).

3. Line 125-128. "Head-prothorax transcriptomes appear well suited for our purpose as they likely encompass components from neuro-endocrine signaling (e.g. linked the JH biosynthesis by the corpora allata), chemical perception and production (antenna and prothorax, respectively), as well as fat body related vitellogenesis". R: This study lacks the ovaries, which also are a presumably key element to explain the queen's fertility. While I think that there are advantages and disadvantages of using the whole body, including the ovaries, versus only the head-prothorax; I think that it could be good someplace in the manuscript, to acknowledge that this study did not take into consideration the ovaries and that this may provide additional insights into the understanding of the trade-off between fertility and longevity.

Author Reply: We agree that ovaries can provide insights into the longevity-fecundity trade-off in queens and that it is worthwhile to do separate investigations. We mention this now explicitly in the text (l. 112-113, 326-329). Studying ovaries alone would be tricky in our species as workers as immatures don't have developed ovaries. Many of the processes linked to fecundity are also happening in the corpora allata and fat body. We covered both with our head-prothorax tissue (though the majority of fat body is in the abdomen). The aim of our study was to investigate fecundity regulation by JH to test the hypothesis of a rewiring along this axis (see also JH manipulation experiment). Therefore, we needed the JH biosynthesis part and thus the corpora allata which is located in the head prothorax region. We feared that whole body tissue would 'flush out' potential signals. Our results also showed that it was not a bad choice to use head-prothorax.

4. Lines 132-133. "Analyses of 28 transcriptomes (14 queens and 14 workers)" R: This number of transcriptomes looks good. However, there is a lack of information on more specific details of the samples to confirm that they can be considered true replicates. For example, are they of similar age and reproductive condition? I

understand that the samples for queen/worker comparison (Data set 1 and Data set 2 A) were collected in the field (data set 1) and the age of the termites is likely unknown, but I could appreciate if the authors could clarify this. On the other hand, in the hormonal manipulation experiments, where the termites were kept in the lab, which were the nutritional conditions used? Can the authors, for example, discard that the collected termites were not experienced nutritional stress?

Author Reply: This is a misunderstanding: the samples for both experiments were obtained in the same way. We recognized that we were not very clear here and we clarified this section of the methods (l. 385-397). All colonies were kept in *Pinus radiata* wood blocks under the same conditions in the lab. With regard to the age of the queen, these ages also slightly varied. Yet, all queens were less than 2 years old since reproducing, fully differentiated and laying eggs. Thus, they were young fertile queens. We added this information to the text (l. 396-397). Despite these variations in our dataset, we found a strong signal for the QCM, indicating the robustness of our results.

5. Lines 174-185. Vg genes: I am extremely curious in knowing more about the function of the different Vg genes in termites. In ants, Vg duplicated and experienced different molecular evolution to acquired caste and behavioral specific functions (e.g., Wurm et al, 2011; Corona et., 2013, Kohlmeler, 2018). What happens in termites? I understand that all the Vg are upregulated in *Cryptotermes* queens. Any clue if they have different functions? Since there is only a true Vg in bees and wasps, termites and ants are the other social insect groups with several Vg genes. I think that it could be very interesting investigate if Vg in termites also underwent subfunctionalization.

Author Reply: Unfortunately, not much is known about the function of different Vg genes in termites. All three Vgs are conventional Vgs, they are the only Vgs found in cockroaches, which include termites, that have a termite-specific Vg duplication (Harrison et al. 2018, unpubl. data of a MS in revision). *Neofem3* seems clearly related to fecundity in *C. secundus* (Weil et al. 2007) as well as in *Cryptotermes cynocephalus* (Weil et al. 2009). Further, all three Vgs are upregulated in queens (and partly in kings) compared to all other castes in the dampwood termite *Zootermopsis nevadensis*, with the strongest upregulation of the *Neofem3* ortholog (Terrapon et al. 2014). These results are all in line with our results. Thus, there is currently not much evidence that implies a subfunctionalisation, though further functional investigations are required. We added more information about the Vgs to the text (l. 164-173).

6. Line 164. JH signal: I agree that high JH epoxidase expression levels suggest increased JH titers, as also have been shown in honey bees and ants. Also, I agree that higher expression of other genes related with JH signal, such as Kruppel is consistent with this view. Please clarify if they are two genes (or isoforms) encoding possible JH epoxidases in *Cryptotermes* as suggested in the figure 3.

Author Reply: Yes, there are two genes that encode JH epoxidase in *C. secundus*. We also state this now more clearly in the text (l. 146-147).

7. Lines 199-206. IIS.
R: I think that here it needs it a little more information on the key elements of the IIS in termites. First, how many insulin receptors are encoded in the *Cryptotermes* genome? There are indeed two, as suggested in figure 3? Second, how many insulin-

like peptides? What happens with the expression of these genes? Unfortunately, the authors do not report caste-specific differences in the expression of these genes that are key to understand if the IIS signal is up-regulated in queens compared with workers as it is suggested. If these genes were not detected (they are normally expressed at very low levels) please clarify. In conclusion, while I agree that higher InR and PDK1 expression in queens suggest a higher insulin signal in queens, I still found the evidence not quite strong without reporting the expression of insulin genes.

Author Reply: Thank you for pointing this out. We agree that we did not provide enough details here. We have now added a new Supplementary Figure 1 (see below) which shows the expression patterns of 'all' IIS genes in queens as revealed from our WGCNA analysis, including all modules. In the figure's legend we also provide additional information on the genes: e.g. there are only two ILPs in *C. secundus*; there are three described *InR* genes in *C. secundus* (*Csec-InR-1*, *Csec-InR-2*, *Csec-InR-3*) (Kremer et al. 2018) and an additional one, *Csec_G15826*, identified in this study. All *InR* genes were not members of the QCM except for *Csec_G15826*. However, they were all associated with queens. *Csec_G15826* was significantly associated with queens and *Csec-InR-2* by trend.

We also added a sentence describing the QCM genes that were differentially expressed and upregulated in queens according to the DEG analysis in the legend of Supplementary Figure 1. For all DEG genes see Supplementary Data 2.

8. Hormonal manipulations. In the first comparison, the authors did not find significant differences in the expression of key genes between the control queens and the precocene treatment. However, when the effect of the solvent was uncoupled, significant differences were apparent in the direction contrary to the honey bee model. Then, If I am right, the solvent masked the effect of precocene. I am not sure if the authors discussed enough why this happened. I think this may be relevant to understand better the results. For example, does JH had a JH-like effect, maybe related to a stress response?

Author Reply: We agree that we should have gone into more detail about the solvent effect and we have added several sentences to the text (l. 104-106, 211-213, 271-278).

It is correct that simple solvent, acetone, seems to have a stress effect in our termites which has also been implied in other studies (e.g. Lengyel et al. 2007). This stress effect may mask the JH effect. Solvent modules seemed to be characterized by stress (e.g. several heat-shock protein/HSP 70 genes were positively associated with solvent treatment, see also Figure 3 heatmap).

Reviewer #3:

1. This study seeks to identify the mechanisms underlying a reversal of life history tradeoffs that has occurred in queens of social insects. The authors use transcriptomes in termite queens to (1) characterize expression modules related to queen function and (2) experimentally test how juvenile hormone influences other components of the molecular pathways associated with life history tradeoffs. They find a module of coexpressed genes that has elements of all the important queen functions, including fecundity, juvenile hormone synthesis, chemical communication, and immunity. They do not, however, find evidence that the set of pathways regulating life history tradeoffs has been "re-wired" in termites. I feel this study addresses an important question with an exciting hypothesis. I appreciated the experimental manipulation included in the study. The methods were mostly rigorous and the paper was well written. In general, I found this to be a compelling study. I do have a couple of outstanding questions, which if addressed, may improve the clarity and impact of the paper.

Author reply: Thank you for these positive words. We appreciated your comments which have helped us to further improve the manuscript.

2. First, the experimental part of the study yielded largely negative results. The hypothesis that the IIS-JH-Vg pathway has been rewired is not supported by the precocene treatments in termites. However, this leaves open the question of how termite queens have had a reversal of life history trade-offs and this is never really explained. For example, the Introduction states that high levels of JH in solitary insects increase fecundity, but also have several life shortening effects. Yet this study reveals that the positive relationship between JH and vg/fecundity seems to be maintained in termites and that queens seem to have an activated JH pathway. So how then is it possible that queens are so long-lived? While this question seems to be driving the research, it does not seem to really be answered by the results. It may be interesting for the authors to speculate a bit here. It might even be interesting to include a fourth panel to Fig 1 illustrating a hypothesized scenario for what might be happening in termite queens. What are we to make of the re-wiring hypothesis?

Author Reply: We would not say that the experimental part yielded largely negative results. We could retrieve the QCM module and showed, for instance, an effect of JH on fecundity - and CHC-related modules. We could also reject the re-wiring hypothesis.

However, we see your point that we are offering no (potential) explanation how termite queens can then overcome the fecundity – longevity trade-off and live so long. Currently, this is still speculative, we think that a re-wiring within some of the axis components (e.g. downstream) could have happened so that it is not the 'major connections/nodes' that are changed, but parts within these modules (e.g. how JH regulates fecundity is not changed but

interactions within the fecundity module interactions are changed). Alternatively (though mutually not exclusive), duplications of some genes, like the *InR* genes, might have opened new options to overcome the trade-off. This is still very speculative. We added a few sentences to the text (l. 350-360).

3. I had some methodological questions about the WGCNA analysis of the precocene experiment. I don't understand why the precocene-treated queens compared to untreated control queens is labeled as the precocene effect. Isn't this really the precocene + solvent effect? It seems to me you would want to isolate the effect of JH by first identifying genes that are differentially expressed between precocene-treated vs control AND precocene-treated vs solvent-treated, but NOT between solvent-treated and control. Then it seems you would want to map these onto the QCM. As it is currently, it seems that WGCNA was applied separately to each treatment group and then modules containing genes from the QCM were identified. I am unclear what the rationale for this is or what it tells us. For example, it is not intuitive to me how to interpret the distribution of genes in the QCM among WGCNA modules in the solvent-treated queens.

Author Reply: It is correct that the precocene effect is actually a combined JH and solvent effect (because JH is applied together with solvent). But we were interested in the 'pure' JH reduction effect. To obtain this, we compared – as you wrote correctly - precocene-treated vs. control AND precocene-treated vs. solvent-treated queens. Additionally, we also tested solvent-treated vs. control queens to see the pure solvent effect (l. 100-106, 208-215).

Note, the WGCNA analyses always includes two treatment groups to test for associations of gene modules with each of the two groups and to identify genes within these modules that significantly associate with this group. Then, we looked at the gene IDs and annotations for each module to assign potential functions (e.g. fecundity-related, CHC related) to them and see how they overlap with the QCM. We recognize this was not clearly stated in the MS and changed the last paragraph of the introduction (l. 100-106).

Therefore, to separate the JH-reduction effect from solvent effect from precocene treated queens, we did WGCNA analysis on precocene-treated queens versus solvent-treated queens. By doing so, we got gene modules with genes that were significantly associated with (significantly associated means in essence, up /down regulated) precocene versus solvent queens (i.e. JH-reduction effect=precocene effect – solvent effect). Similarly, by doing WGCNA analysis on solvent-treated queens and untreated control queens, we were able to detect the gene modules with genes that were significantly associated with solvent queens versus untreated queens (i.e. solvent effect). Finally, by doing WGCNA analysis on precocene-treated queens and untreated control queens, we were able to detect the gene modules with genes that were significantly associated with precocene queens versus untreated queens (i.e. precocene effect=JH-reduction effect + solvent effect). For each of these three analyses, we then looked at the gene IDs and annotations to see which functions the modules were related to.

4. Further, I am confused about how the WGCNA was done. In the section 'Disentangling JH reduction from handling artifacts', it sounds as though perhaps only genes that are differentially expressed between precocene-treated and solvent-treated queens were used in the WGCNA, but this seems to contradict what is stated on L136 and L475.

Author Reply: Sorry for the confusion. We always used all expressed genes that passed quality filtering, not only DEGs for all WGCNA analyses. In addition, we did DEG analyses with DeSeq2, independent from the WGCNA analysis. Each of the corresponding sections starts with the DEG results before the WGCNA results. To make this clearer, we changed the last paragraph of the introduction (l. 96-100).

- I also found myself questioning the choice of tissue used for transcriptomics. Some of this justification came at the very end, but it would be helpful to this reader if that justification came earlier. Fig. 4 goes so far as to hypothesize the functional pathways of the newly discovered QCM, including purported function in the abdomen. I am not clear on how the coexpression modules detected in the head/prothorax can inform us about molecular pathways operating in other tissues of the body. L176 points out that vitellogenins are produced in the fat body and carried through the hemolymph to the ovary for oogenesis. Does this include fat body in the head?

Author Reply: We now explain tissue choice at the end of the introduction (l. 109-113). As we used prothorax combined with head, we obtained fat body from the prothorax, as well as hemolymph. This is likely where the vitellogenin signal comes from. To make this more explicit, we changed the text and added the caveat of missed signals (l. 112-113, 320-327) and also modified Figure 4 (see below).

- Some specific questions about methods. L393: How many colonies were used to generate the queens in this study? Were any of them related? If so, I would think this would need to be accounted for in statistical analyses.

Author Reply: Thanks for this question, we were not specific enough here. In total we used more than 80 colonies to generate experimental colonies. The queens and workers used in this study all came from different (stock) colonies. Thus, the data represent independent biological replicates. We added this information now to the text (l. 391-397).

7. L294: "identified four JH-effect modules of co-expressed genes related to the QCM". Does this just mean genes belonging to the QCM were found in these modules? Or was there significant overlap among the modules? Attributes to the modules are being stated as they relate to the function of just a few genes. For example, JH-effect module *orangered1* is described as containing 7 genes related to fecundity. But is this a significant enrichment? How many genes are in this module? This is a recurring issue throughout this section.

Author Reply: We indeed also added 'functional names' to modules after most striking genes that significantly associated with them. We did this to improve readability as there were really striking gene patterns in these modules (see Supplementary Data 3). So it is not that we named the modules after 1-2 genes. It would have been nice to do a test of enrichment for these functions – and we thought about it - but this is not possible/reasonable: We would have to classify genes. This might be possible to do for the genes from the modules but impossible for the background (i.e. all expressed genes in the study). We did enrichment analyses like KEGG and GO (see Supplementary Data 5 & 6). Yet they were not very helpful for our purpose to link to the QCM: For instance, the categories we are looking at do not have GO terms (e.g. chemical communication) or are not well annotated (e.g. this related with JH biosynthesis and signalling). We added some details to address these points (l. 219-225). The exact number of genes and gene lists within each module are also provided and can be found in Supplementary Data 3.

8. L 462: It sounds like just the DEGs for each comparison is included in the WGCNA, but L136 and L475 seems to indicate all genes were used. If the latter, then what does the comparison refer to here?

Author Reply: The WGCNA was based on all genes, not only DEGs. This is what is recommended by Langfelder & Horvath (2008). Limiting this analysis to DEGs would mean considerable loss of information. The comparison here refers to the calculation of gene-trait and module-trait association coefficients in WGCNA, not in DEG analysis, with 'trait' referring to 'caste' (queen vs. worker) in Experiment 1 and 'treatment groups' (e.g. precocene vs. control) in Experiment 2. In short, it compares the expression of a certain gene - or the expression of the first principal component of a gene cluster - between different trait groups. For example, if the expression of gene A increases significantly when comparing treatment 1 to treatment 2, it indicates that gene A is positively associated with treatment 2. Thus, to obtain the modules, a co-expression analysis of genes is performed to cluster genes with similar expression patterns into a module. Please, see l. 454-490 for further information.

9. L542: Where were the results of the KEGG pathway enrichment tests presented?

Author Reply: They were not presented in the main text but only in Supplementary Data 5. These analyses did not provide many new insights. Therefore, we had not mentioned them in the text. Yet, for reasons of completeness, we added them as they might be helpful for other researchers and upcoming studies and we now refer to them in the main text (l. 136, 224-225).

References

Harrison M.C., Jongepier E., Robertson H.M., Arning N., Bitard-Feildel T. et al. (2018). Hemimetabolous genomes reveal molecular basis of termite eusociality. *Nat. Ecol. Evol.*, 2, 557-566.

Hoffmann K. & Korb J. (2011). Is there conflict over direct reproduction in lower termite colonies? *Anim. Behav.*, 81, 265-274.

Hoffmann K., Gowin J., Hartfelder K. & Korb J. (2014). The scent of royalty: a P450 gene signals reproductive status in a social insect. *Mol. Biol. Evol.*, 31, 2689-2696.

Keller L. & Genoud M. (1997). Extraordinary lifespans in ants: a test of evolutionary theories of ageing. *Nature*, 389, 958-960.

Kremer L.P.M., Korb J. & Bornberg-Bauer E. (2018). Reconstructed evolution of insulin receptors in insects reveals duplications in early insects and cockroaches. *J. Exp. Zool. B. Mol. Dev. Evol.*, 330, 305-311.

Langfelder P. & Horvath S. (2008). WGCNA: an R package for weighted correlation network analysis. *Bioinformatics*, 9, 559. <https://doi.org/10.1186/1471-2105-9-559>

Lengyel F., Westerlund S.A. & Kaib M. (2007). Juvenile hormone III influences task-specific cuticular hydrocarbon profile changes in the ant *Myrmicaria eumenoides*. *J. Chem. Ecol.*, 33, 167-181.

Terrapon N., Li C., Robertson H.M., Ji L., Meng X. et al. (2014). Molecular traces of alternative social organization in a termite genome. *Nature Communications*, 5, 3636.

Weil T., Korb J. & Rehli M. (2009). Comparison of queen-specific gene expression in related lower termite species. *Mol. Biol. Evol.*, 26, 1841-1850.

Weil T., Rehli M. & Korb J. (2007). Molecular basis for the reproductive division of labour in a lower termite. *BMC Genomics*, 8, 198.

REVIEWERS' COMMENTS:

Reviewer #1 (Remarks to the Author):

Modifications have been done, no more comments on my side

Reviewer #2 (Remarks to the Author):

The authors answered my questions and corrected the manuscript as suggested.

Reviewer #3 (Remarks to the Author):

The authors have satisfactorily addressed all of the points raised in my review of an earlier version of this manuscript. I believe the clarity has been improved, and this will be an important study. I recommend it be accepted for publication.